# Effects of Rehabilitation on Perineural Nets and Synaptic Plasticity Following Spinal Cord Transection

**DOI:** 10.3390/brainsci10110824

**Published:** 2020-11-06

**Authors:** Yazi D. Al’joboori, V. Reggie Edgerton, Ronaldo M. Ichiyama

**Affiliations:** 1Faculty of Biological Sciences, School of Biomedical Sciences, University of Leeds, Leeds LS2 9JT, UK; y.aljoboori@ucl.ac.uk; 2Physiological Science, Neurobiology and Brain Research Institute, University of California, Los Angeles, CA 90095, USA; vre@ucla.edu

**Keywords:** spinal cord injury, rehabilitation, chondroitin sulfate proteoglycans, neuromodulation, gamma motoneuron

## Abstract

Epidural electrical stimulation (ES) of the lumbar spinal cord combined with daily locomotor training has been demonstrated to enhance stepping ability after complete spinal transection in rodents and clinically complete spinal injuries in humans. Although functional gain is observed, plasticity mechanisms associated with such recovery remain mostly unclear. Here, we investigated how ES and locomotor training affected expression of chondroitin sulfate proteoglycans (CSPG), perineuronal nets (PNN), and synaptic plasticity on spinal motoneurons. To test this, adult rats received a complete spinal transection (T9–T10) followed by daily locomotor training performed under ES with administration of quipazine (a serotonin (5-HT) agonist) starting 7 days post-injury (dpi). Excitatory and inhibitory synaptic changes were examined at 7, 21, and 67 dpi in addition to PNN and CSPG expression. The total amount of CSPG expression significantly increased with time after injury, with no effect of training. An interesting finding was that γ-motoneurons did not express PNNs, whereas α-motoneurons demonstrated well-defined PNNs. This remarkable difference is reflected in the greater extent of synaptic changes observed in γ-motoneurons compared to α-motoneurons. A medium negative correlation between CSPG expression and changes in putative synapses around α-motoneurons was found, but no correlation was identified for γ-motoneurons. These results suggest that modulation of γ-motoneuron activity is an important mechanism associated with functional recovery induced by locomotor training under ES after a complete spinal transection.

## 1. Introduction

Rehabilitation, such as locomotor training, remains one of the only clinical interventions to promote functional sensorimotor recovery after a severe spinal cord injury (SCI). We originally demonstrated that after a complete spinal cord transection, full weight-bearing coordinated stepping can be elicited with a combination of epidural electrical stimulation (ES) and serotonergic (5-HT) agonists [1,2]. This ability was subsequently enhanced with daily locomotor training with further improvements in step kinematics and other locomotion control parameters [3,4,5]. These beneficial effects of ES following severe SCIs have additionally been shown to translate to humans with recovery of voluntary contraction of lower limb muscles, standing ability [6,7], and, most recently, walking [8,9]. Nonetheless, the mechanisms underlying such functional recovery with ES and locomotor training remain largely unknown.

Inhibitory molecules present in the spinal cord after an injury restrict axonal plasticity limiting the ability of the spinal cord to remodel [10,11]. Perineuronal nets (PNNs) in the CNS are formed of extracellular matrix (ECM) molecules, chondroitin sulphate proteoglycans (CSPGs). Secreted CSPGs from neurons and glial cells assemble extracellularly on the surface of cell somata and proximal dendrites of certain classes of neurons to form PNNs [12,13]. PNNs are thought to play both protective and inhibitory roles in the CNS. Several studies investigating the nature of PNNs in relation to plasticity have adopted the use of changes in fluorescence intensity of the *Wisteria floribunda* agglutinin (WFA). WFA is a plant lectin which binds to N-acetylgalactosamine residues that form glycan chains of the chondroitin sulphate proteoglycan (CSPGs) aggregate to form PNNs. Evidence strongly suggests that PNNs are associated with activity-dependent synaptic plasticity in the brain during development [14]. Specifically, the expression of WFA+ive PNNs is decreased or completely absent before the critical period of visual cortex development, when synapse formation is taking place. Following closure of the critical period, PNNs are then initially formed and are upregulated through maturity where they are dynamically maintained [14]. Those observations indicated a critical role of PNNs in consolidating functional circuits during development. This phenomenon was further demonstrated by application of chondroitinase ABC (ChABC), an enzyme that digests CSPG glycosaminoglycan (GAG) side chains following a monocular visual deprivation study. ChABC was shown to reactivate ocular dominance following monocular deprivation by digesting PNNs and promoting plasticity in the CNS. This effect has been applied in relation to other functional deficits where CSPGs have been shown to impede functional recovery following an injury [15,16]. The role of PNNs in rehabilitation after a spinal cord injury is much less studied. Wang et al. [17] demonstrated that treatment with ChABC starting four weeks after an incomplete lesion resulted in upregulation of PNNs around motoneurons in rats receiving rehabilitation compared to no rehabilitation. However, it is still unclear how rehabilitation alone (without ChABC treatment) modulates the expression of PNNs in a severe injury model and how this regulation affects synaptic plasticity.

This study was conducted to evaluate the effects of locomotor training and ES on the regulation of PNNs in the spinal cord following a complete spinal transection. The relationship between synaptic remodeling and expression of CSPGs was evaluated in spinally transected non-trained and trained rats at different time points after a lesion. Both inhibitory and excitatory boutons on ventral horn α- and γ-motoneurons (α- and γ-MNs) were investigated, as we had previously demonstrated differential effects of an injury and rehabilitation on the two subtypes of motoneurons [18,19]. Our results showed that significantly greater synaptic remodeling was observed for glutamatergic boutons apposing γ-MNs (compared to α-MNs). Interestingly, γ-MNs also displayed a lack of PNNs.

## 2. Materials and Methods

All experimental procedures complied with the guidelines of the National Institute of Health Guide for the care and use of laboratory animals, the University of California, the University of Leeds, and the United Kingdom Animals Act 1996. This project was performed under UK Home Office project license 70/8085 obtained on July 14, 2014 (animal experiments completed in 2016). Adult female Sprague–Dawley rats (250 g) were used in all experiments, which were housed individually exposed to a 12 h light/dark cycle with food and water available *ad libitum*.

### 2.1. SCI and Epidural Implantation

Full details of methods have been previously reported [1,2]. Briefly, animals were anesthetized with isoflurane (5% in O_2_) and three partial laminectomies were performed over spinal segments T9/T10, L2, and S1, which were identified using anatomical landmarks to identify vertebral segments. At T9/T10, the spinal cord was completely transected using fine scissors; both stumps were lifted to ensure a complete transection which was verified by the second surgeon. Gel foam was inserted into the gap to prevent reconnection of cut ends of the spinal cord. Implantation of epidural stimulation electrodes was then performed at L2 and S1 (Figure 1). A Teflon-coated stainless steel wire (Cooner Wire Company, Chatsworth, CA, USA) was passed under the spinous processes above the dura mater between partial laminectomy sites. After an opening on the Teflon coating was created, the wire was sutured to the dura mater at the midline using a 9.0 ethilon suture. This process was repeated for the L2 and S1 implantation sites. The wires were connected to an Amphenol connector (Omnetics Connector Corporation, Minneapolis, MN, USA)) cemented to the skull.

### 2.2. Locomotor Training

As mentioned above, the details of parameters for locomotor training have been previously published [1]. One week post-injury, the animals were assigned to treatment groups: 7 days post-injury (dpi; *n* = 7); 21 dpi, non-trained (*n* = 5); 21 dpi, trained (*n* = 5); 67 dpi, non-trained (*n* = 10); and 67 dpi, trained (*n* = 10). Not all animals were used in all histological experiments. Trained animals received an i.p. injection of quipazine (0.3 mg/kg) 15 min before each training session. Epidural stimulation was performed using a Grass stimulator (S88X stimulator; Astro-Med^®^, Inc., Grass Instruments, Middleton, WA, USA) delivering continuous biphasic stimulation at 40 Hz with 200 µs rectangular pulses. Voltage was controlled via an isolation unit (SIU-V isolation unit; Astro-Med^®^, Inc., Grass Instruments, Middleton, WA, USA) for individual rats to optimize stepping. Training under quipazine and epidural stimulation commenced 7 dpi, was performed daily (5 days/week) for 30 min for 8 weeks. Non-trained rats did not receive any treatment (Figure 1).

We have previously demonstrated that after 8 weeks of training under epidural stimulation and quipazine, rats are able to consistently step for 30 min with step kinematics which are indistinguishable from neurologically intact rats [1,3]. We have also demonstrated that at 21 dpi, epidural stimulation is only able to elicit rudimentary locomotor-like movements with poor step kinematics [20]. Consistent weight-supported stepping is only observed under epidural stimulation after 6 weeks post-injury [20]. Therefore, 21 dpi will demonstrate the early stages of locomotor training. At 7 dpi, completely transected rats show no ability to support their weight or step. Each one of the time points chosen for this study will demonstrate a completely unique stage in the recovery of locomotor function after a complete transection and training.

### 2.3. Tissue Preparation and Immunohistochemistry

After each appropriate period, rats received a lethal dose of pentobarbital (200 mg/kg) and were transcardially perfused with 0.1 M phosphate buffer saline (PBS) following loss of all withdrawal reflexes. Fixation was performed with 4% paraformaldehyde (PFA), which was followed by a final wash with PBS. Spinal cords were grossly dissected with the vertebral column and post-fixed overnight in the same fixative. After careful dissection from the bony vertebral column encasing, spinal cords were cryoprotected in 30% sucrose in PBS for at least 3 days. The identified spinal segments were embedded and frozen in the optimum cutting temperature (OCT) medium and later sectioned into 25 µm coronal sections using a cryostat (Leica CM 1850).

Free-floating sections were washed (three times, 10 min per wash) in PBS. Sections were blocked for 2 h in 3% normal donkey serum (NDS, Sigma with PBS containing 0.2% Triton X-100) at room temperature. Sections were immediately incubated in primary antibodies (Table 1) overnight at 4 °C, triple or double stained for specific primary antibodies. Sections were then washed in PBS (three times, 10 min per wash) before blocking again for 1 h in 3% NDS. To visualize CSPGs and PNNs, blocked sections were incubated for 2 h at room temperature in biotinylated *Wisteria floribunda* agglutinin (WFA, L1516-2MG, Merck Life Science UK Limited, Gillingham, United Kingdom, 1:500). Sections were then washed again in PBS (three times, 10 min per wash) and transferred into species-appropriate secondary antibodies (Table 1). For biotinylated antibodies, sections were incubated in Pacific Blue streptavidin (Invitrogen, 1:500) for 2 h. After a final PBS wash, sections were mounted onto slides and coverslipped with the Vectashield mounting medium (Vector Laboratories Inc, Burlingame, CA, USA).

### 2.4. Image Analysis

For synaptic boutons counting, images were obtained using an inverted Zeiss LSM 510 META Axiovert 200M confocal microscope with an oil immersion 63× objective lens. In order to visualize γ-aminobutyric acid (GABA)ergic and glutamatergic synaptic boutons apposing α- and γ-MNs, triple immunofluorescence-stained sections were analyzed for choline acetyl transferase (ChAT), neuronal nuclei (NeuN), and vesicular glutamate transporter 2 (VGluT2) or ChAT, NeuN, and vesicular GABA transporter (VGAT). Three random L5 sections from each animal were scanned and pictures from every single motoneuron (ChAT+ neurons in the ventral horn) displaying a nucleus were captured for analysis. The mean diameter and perimeter of motoneurons were measured using the LSM 510 META software. The number of boutons was normalized to the motoneuron perimeter.

For CSPG and PNN quantification, WFA was analyzed in 3 random L5 sections from each rat. WFA intensity was measured in tiled 20× scans of the entire spinal section using SimplePCI (Hamamatsu). WFA thickness around motoneurons was measured in the images captured using a 40× oil immersion lens. Mean thickness was calculated by subtracting the radius calculated from the area encompassing both motoneurons and PNNs from the radius calculated from the MN area. Ventral horn motoneurons (ChAT+) containing perineuronal nets were readily identified. Perineuronal nets were visible as tightly and intensely stained areas around some neurons, clearly distinguishable from the more diffused CSPG staining of the extracellular matrix.

### 2.5. Statistical Analysis

All data are displayed as the mean ± SEM in figures and tables. To determine differences between intact, 7 days post-injury (7d), 21 days post-injury non-trained (21dNT), 21 days post-injury trained (21dTR), 67 days post-injury non-trained (67dNT), and 67 days post-injury trained rats (67dTR), a one-way analysis of variance (ANOVA) with Fisher’s Least Significant Difference (LSD) post-hoc tests was performed. For specific training or time-after-injury effects, a 2 × 2 factorial ANOVA was used and a 2 × 3 × 2 factorial ANOVA was used to test for the effects of the motoneuron type (α or γ-MN), days post injury (7, 21, or 67 dpi), and training (trained or non-trained). Finally, to determine relationships between synaptic plasticity and CSPG expression, we regressed the percent change of boutons (VGluT2 and VGAT) in comparison to intact rats onto the amount of CSPG expression (WFA staining) in the spinal cord.

## 3. Results

### 3.1. Increased Expression of CSPGs with Time Following Injury

The expression of CSPGs in the lumbar spinal cord significantly increased with time after an injury (*F* = 4.52; *p* < 0.05) with a significant simple effect for days (*F* = 10.67; *p* < 0.01), but not for training (*F* = 2.55, *p* > 0.05) or their interaction (*F* = 2.43; *p* > 0.05) (Figure 2A,B). LSD post-hoc tests indicated that 67dTR and 67dNT had a significantly higher expression of CSPGs compared to intact, 7d, and 21dNT rats. These results indicate that CSPG expression is increased with time after an injury, but daily locomotor training did not have an effect of CSPG expression (Figure 2B).

### 3.2. Number of PNNs

Given the results above, we focused our further analysis on the 67d groups. We specifically investigated changes in PNNs by counting the number of neurons expressing them in the entire grey matter and measuring PNN thickness and intensity of labelling around motoneurons. The number of neurons expressing PNNs in the lumbar spinal cord did not differ significantly after a complete thoracic spinal transection or after training (*F* = 0.38, *p* = 0.68; Figure 2C). There was, however, a main effect for laminae (*F* = 66.57, *p* < 0.001). Laminae I, II, and X had significantly fewer PNNs compared to all other laminae (Figure 2C). The interaction between groups and laminae was not statistically significant (*F* = 0.77, *p* = 0.73), indicating that neither the injury nor locomotor training had an effect on the number of neurons expressing PNNs.

### 3.3. PNN Expression in Motoneurons

In order to identify α-MNs, we used both ChAT and NeuN as markers: ChAT+ NeuN+ neurons in lamina IX are α-MNs, whereas ChAT+ NeuN– neurons in lamina IX are γ-MNs [21,22]. We observed that all NeuN+ ChAT+ neurons (putative α-MNs) in lamina IX expressed PNNs and therefore we investigated this population further. Surprisingly, using these markers, we also determined that unlike α-MNs, γ-MNs do not express PNNs (Figure 2D–G). This was also observed in spinally transected rats and in trained rats.

For α-MNs, training did not affect either thickness of PNNs (*t* = −1.669, *p* = 0.146) or intensity of staining (*t* = −0.847, *p* = 0.437), as no significant differences were observed when comparing non-trained and trained rats (Table 2), which mirrors the results of general CSPG expression in the ventral horn for rats in 67dNT and 67dTR groups (Figure 2B).

### 3.4. Absolute Number of Synaptic Boutons

Next, we investigated how excitatory (glutamatergic, VGluT2, Figure 2D–G) and inhibitory (GABAergic and glycinergic, VGAT, Figure 3D–G) putative synapses changed after an injury and locomotor training around both α- and γ-MNs. The absolute number of VGluT2 boutons normalized to 100 µm of the membrane surface apposing α-MNs significantly changed with an injury and training (*F* = 3.66, *p* < 0.05; Figure 3A), whereas there were no significant differences in γ-MNs (*F* = 1.92, *p* = 0.15, Figure 3B). The changes in α-MNs were not linear. Absolute numbers of VGluT2 boutons significantly increased with time after an injury with significant differences between intact, 7, 67dNT, and 67dTR rats (*p* < 0.05). Interestingly, 21dTR values were significantly lower than for 7d, 21dNT, 67dNT, and 67dTR rats (*p* < 0.05; Figure 3A).

For VGAT boutons, there were no significant differences in the absolute number of boutons apposing α-MNs (*F* = 0.90, *p* = 0.50; Figure 4A) or γ-MNs (*F* = 0.51, *p* = 0.76, Figure 4B) between conditions.

### 3.5. Synaptic Remodeling after an Injury and Training

In order to better determine how an injury and locomotor training changed synaptic boutons apposing motoneurons, we calculated the percent change in bouton numbers (normalized to the 100 µm membrane length) in relation to neurologically intact rats. This ratio indicates how the transection injury and locomotor training influenced the changes in synaptic apposition in lumbar motoneurons. For VGluT2 boutons, the percent change from intact controls was significantly greater in γ-MNs compared to α-MNs (*F* = 7.27, *p* < 0.05; Figure 3C). We also observed a significant interaction between motoneuron and training group (*F* = 4.51, *p* < 0.05) indicating that training resulted in greater plasticity of VGluT2 boutons in close apposition with γ-MNs, but did not have the same effect on α-MNs. No other factors or interactions were significant.

For VGAT, there were no significant main effects for motoneurons (*F* = 2.51, *p* = 0.12) or groups (*F* = 0.91, *p* = 0.48). Similarly, there were no effects of training (*F* = 0.19, *p* = 0.66) and no interactions between any factors (Figure 4C).

### 3.6. Relationship between Synaptic Changes and CSPG

In order to determine the relationship between CSPG expression and changes in synaptic composition in motoneurons after an injury and locomotor training, we regressed the percent change of number of synaptic boutons per 100 µm of membrane length onto WFA intensities. The latter choice is due to the fact that γ-MNs do not express perineuronal nets (Figure 2D–G, arrow) and therefore, WFA intensity within the grey matter was used instead of specific measurements of the perineuronal net thickness.

For α-MNs, a moderate negative correlation was observed when data for 67 dpi were analyzed with VGluT2 (*R*^2^ = 0.052, partial correlation: −0.288; Figure 5A) and VGAT (*R*^2^ = 0.4285, partial correlation: −0.655; Figure 5C). These results indicate that at 67 dpi, synaptic plasticity was highest in α-MNs expressing the lowest levels of CSPGs.

For γ-MNs, no correlations were observed for either VGluT2 (*R*^2^ = 0.0064; partial correlation: −0.08; Figure 5B) or VGAT (*R*^2^ = 0.0143; partial correlation: −0.092; Figure 5D). These results indicate that synaptic changes in γ-MNs are independent of levels of CSPG expression.

## 4. Discussion and Conclusions

Spinal cord injury and locomotor training under ES resulted in significant restructuring of spinal networks caudal to the level of injury. Here, we demonstrate for the first time the relationship between a plasticity-inhibiting mechanism (CSPGs and PNNs) and changes in synaptic composition of boutons in close apposition to α- and γ-MNs in the spinal cord following a complete transection and locomotor training under ES. Our main findings indicate that, regarding absolute numbers, both injury and training modulated significant changes in glutamatergic boutons apposing α-MNs, but no changes were observed for GABAergic and glycinergic boutons. However, when the percent change in relation to intact levels was analyzed, a greater synaptic change in VGluT2 boutons was observed in γ-MNs after a spinal transection and locomotor training than in α-MNs. CSPG expression significantly increased with time after an injury. However, training did not have an effect on the number of neurons expressing PNNs in the spinal grey matter or the thickness of PNNs around motoneurons. Interestingly, we did observe that γ-MNs do not express PNNs as α-MNs do. Finally, a negative correlation between the percent change in both VGluT2 and VGAT boutons and WFA staining was observed for α-MNs at 67 dpi. Γ-MNs did not show any relationship between percent changes in boutons and WFA staining.

Both we and others have previously reported in detail the recovery of stepping ability after 6–8 weeks of daily locomotor training, quipazine, and ES after a severe SCI [1,3,5]. Here, we compared spinal cords at 7, 21, and 67 dpi. At 7 dpi, a mid-thoracic complete transection resulted in complete paralysis of the hindlimbs and no locomotor movements were observed. 21 dpi corresponds to two weeks of locomotor training. Both we and others have previously reported that weight-supported stepping movements are not observed until 3 weeks following initiation of locomotor training under ES and quipazine [20]. Therefore, it was not possible to compare step kinematics between different time periods following an injury, because, in effect, 7 and 21 dpi groups did not step independently even under ES and quipazine.

### 4.1. Synaptic Remodeling after an Injury and Training

We have previously demonstrated that γ-MNs are more susceptible than α-MNs to synaptic changes after a complete spinal transection performed at post-natal day 5 (P5) and daily locomotor training [18,19]. Even though in the present study the transection was performed after the animals had fully matured and training was done under epidural stimulation and quipazine administration, our results on γ-MN plasticity were consistent with our previous studies. Interestingly, in the present adult model of an injury, the changes in inhibitory putative synapses were not significant as they were for the neonatal injury model [19]. Developmental issues may account for these differences, as most supraspinal projections to the lumbar spinal cord are either immature or not present at P5 [23,24]. Chakrabarty and Martin [25] have demonstrated that co-development of muscle afferents and descending projections, specifically from the corticospinal tract, is critical to maintain the correct balance between excitation and inhibition in the spinal cord. It is likely that the effects of a complete spinal transection on synaptic apposition of motoneurons differ significantly between a neonatal and an adult transection model [26,27], potentially accounting for the discrepancies observed in the current study.

### 4.2. CSPG Expression and PNNs after an Injury and Training

Upregulation of CSPGs around the lesion site has been repeatedly demonstrated [11,28,29,30] and treatment with ChABC targeting the glial scar has been proven effective in decreasing the scar allowing axonal sprouting in incomplete lesion models [31]. Investigations of PNN regulation caudal to the lesion, however, are much scarcer. We have previously reported an upregulation of CSPGs and PNN expression just caudal to the lesion site (200 µm) even several weeks after the initial insult [17]. Here, we report progressive upregulation of CSPG expression many segments caudal to the lesion, with the highest levels present at the latest time point analyzed (67 dpi). It was interesting to note that CSPG expression in the ventral horn caudal to the lesion was not significantly increased at 7 or 21 dpi, unlike what is seen in the formation of the lesion scar [32].

Although there was a trend for the CSPG expression to be increased in the trained group at 21 dpi, this difference was not significant at either 21 or 67 dpi. At 67 dpi in this complete transection model, locomotor training did not have a significant effect on CSPG expression or on PNN thickness. These results are in contrast to our previous observations in a cervical dorsal column lesion where rehabilitation increased levels of CSPGs beyond that observed in the post-injury group without rehabilitation [17]. Recently, we have also observed that exercise training (voluntary running wheel) significantly increased the expression of CSPGs in the lumbar spinal cord of neurologically intact rats compared to sedentary rats [33]. Those results suggest that increased peripheral activity (exercise/rehabilitation) results in increased expression of CSPGs and PNNs in the corresponding spinal circuits. In the current study, however, a complete lack of descending and/or ascending connections within the lumbar cord may have played a role in the absence of such differences between trained and non-trained groups. Even though rats are able to locomote under ES, they remain paralyzed without ES in their home cages. It is plausible that 30 min of activity per day were not sufficient to cause statistically significant increases in expression of CSPGs. The interaction between the rehabilitation modality and the severity of an injury remains unclear.

### 4.3. Synaptic Remodeling and CSPGs

It was interesting to observe that there was a distinct dichotomy in the expression of PNNs around α-MNs and γ-MNs. All α-MNs, but no γ-MNs expressed PNNs, which was not changed after an injury or training. We demonstrated a clear relationship between CSPG expression and the amount of synaptic remodeling in α-MNs, but no such relationship existed in γ-MNs. Additionally, γ-MNs demonstrated the largest relative amount of synaptic remodeling. These results combined are in direct agreement with the current view that expression of PNNs is correlated with a lower plasticity potential as observed in ocular dominance experiments [14,34] and more recently in hippocampus plasticity and learning [35]. The fact that γ-MNs did not express PNNs and also showed the greatest percent changes in synaptic plasticity further corroborates this hypothesis. There is compelling evidence that muscle spindle afferents are necessary for the recovery of stepping afforded by daily locomotor training under ES following an SCI [20,36,37]. Γ-MNs innervating the intrafusal fibers of muscle spindles could play a critical role in such a recovery. It will be important to investigate this further in the future.

## 5. Clinical Implications

Given that synaptic plasticity was negatively correlated with CSPG expression in α-MNs and, as previously demonstrated, with the recovery of the locomotor function [19], facilitating synaptic plasticity by manipulating CSPG and PNN expression after an injury may improve the effects of rehabilitation on the functional recovery. Designing clinical intervention strategies aimed at enhancing plasticity within the spinal cord should be undertaken with the aim to make appropriate useful reconnections. As such, the use of the chondroitinase ABC (ChABC) enzyme has been shown to effectively enhance the effects of rehabilitation of forepaw reaching movements after a cervical dorsal column lesion in rats [38]. However, for the recovery of locomotion after more severe lesions, the effects of combinatorial interventions have been shown to depend on the timing of delivery of each treatment [39,40,41,42]. Indeed, when two plasticity-enhancing interventions were combined (anti-Nogo-A antibody and ChABC), the timing of delivery for each component was also critical for successful functional recovery [43]. Therefore, it is likely that when combining ChABC and locomotor-enhancing rehabilitation, such considerations should also be explored.

## Figures and Tables

**Figure 1 brainsci-10-00824-f001:**
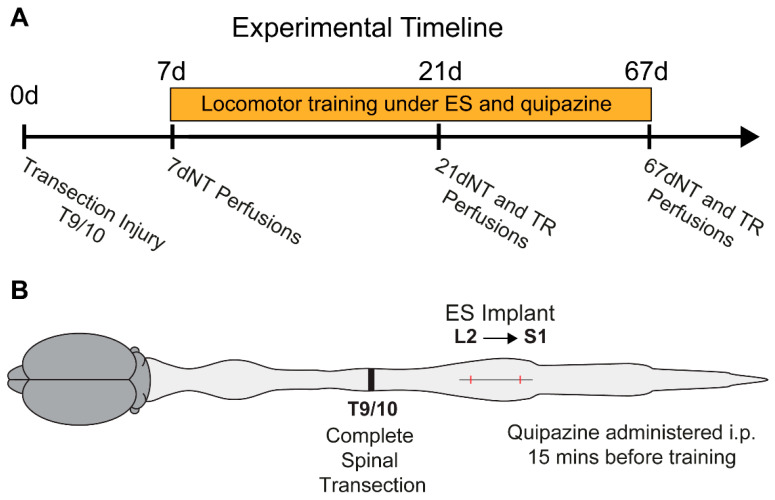
Schematic of the experimental timeline, injury, and electrode implantation. (**A**) Daily locomotor training under epidural electrical stimulation (L2–S1, 40 Hz, 30 min/day) and quipazine (i.p., 0.3 mg/kg) started one week following a transection injury. Different groups were transcardially perfused at 7, 21, or 67 days post-injury. (**B**) Schematic of the location of the complete transection and implantation of epidural electrodes. 7dNT = 7 days, non-trained; 21dNT = 21 days, non-trained; TR = trained; 67dNT = 67 days, non-trained.

**Figure 2 brainsci-10-00824-f002:**
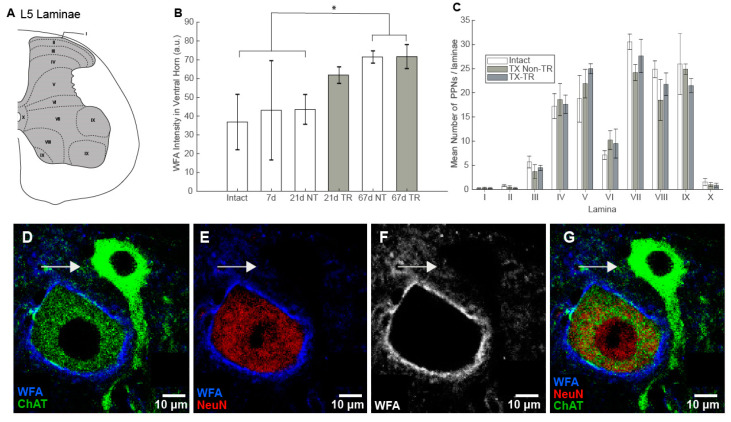
Schematic of laminar distribution at lumbar segment 5 (L5). (**A**) Intensity of WFA staining in the spinal cord grey matter. (**B**) A significant difference was observed between 67dpi groups and intact, 7d, and 21dNT groups. (**C**) Number of PNNs in the intact and 67dpi groups. Laminae I, II, III, and X demonstrated the lowest number of PNNs. However, there were no significant differences between groups for any lamina. Γ-MNs do not express PNNs: (**D**–**G**) motoneurons in the ventral grey matter labeled with ChAT (green); only α-MNs co-labeled with NeuN (red), whereas γ-MNs (arrow) are NeuN-. (**E**,**F**) α-MNs are clearly enveloped by a perineuronal net (WFA), but γ-MNs do not express any organized perineuronal net (F, image shown in black and white for clear visualization). * = *p* < 0.05. TX Non TR = Transected, non-trained; TX TR = Transected trained.

**Figure 3 brainsci-10-00824-f003:**
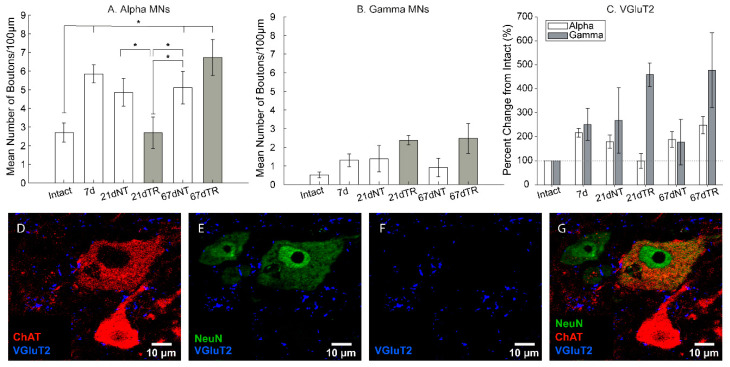
Glutamatergic boutons were detected with the antibody against VGluT2. (**A**) In α-MNs, significant differences in the absolute number of boutons were observed between 67 dpi groups and intact, 7d, and 21dNT groups. (**B**) In γ-MNs, no significant differences in the absolute number of VGluT2 boutons were observed. In (**C**), the percent change of VGluT2 boutons was significantly higher in γ-MNs. (**D**) Motoneurons were identified as ChAT+ (red) cells in the ventral horn. (**E**) NeuN (green) is only expressed in α-motoneurons and other interneurons. (**F**). VGluT2 boutons (blue) were identified in close apposition to both γ- and α-motoneurons. (**G**) Combined triple filters. * = *p* < 0.05.

**Figure 4 brainsci-10-00824-f004:**
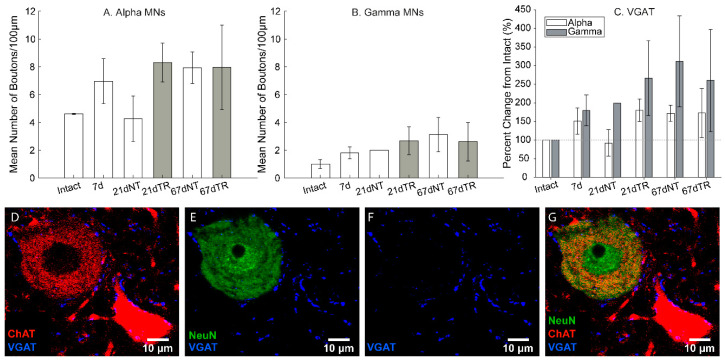
Intraspinal GABAergic/glycinergic boutons were detected with the antibody against VGAT. No significant differences in the absolute number of VGAT boutons were observed in α-MNs (**A**) or γ-MNs (**B**). In (**C**), the percent change of VGAT boutons showed no significant differences between α- and γ-MNs. Motoneurons were identified as ChAT+ (red) cells in the ventral horn. (**D**) NeuN (green) is only expressed in α-motoneurons and other interneurons. (**E**). VGAT boutons (blue) were identified in close apposition to both γ- and α-motoneurons. (**F**). (**G**) Combined triple filters.

**Figure 5 brainsci-10-00824-f005:**
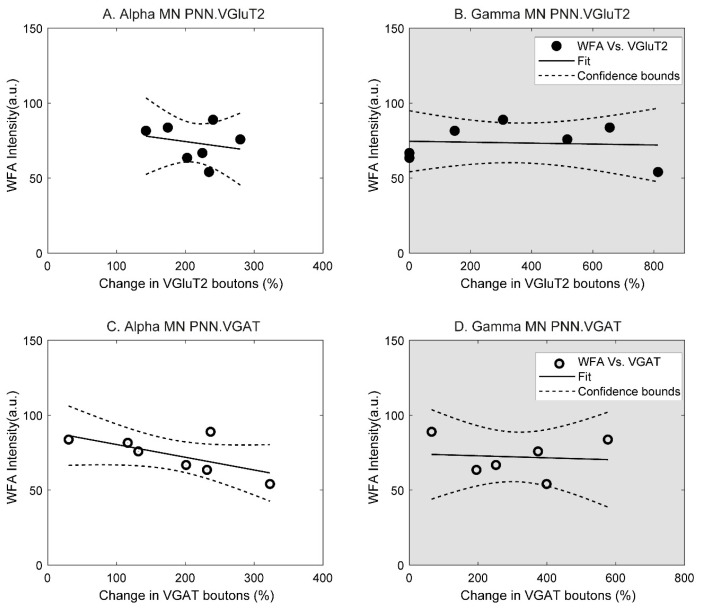
In α-MNs, a negative correlation between WFA intensity and percent change in synaptic boutons develops after 67 days post-injury for VGluT2 (**A**) and VGAT (**C**). In γ-MNs, no relationship between WFA intensity and changes in synaptic boutons was observed for either VGluT2 (**B**) or VGAT (**D**).

**Table 1 brainsci-10-00824-t001:** List of antibodies.

Antibodies	Supplier	Species	Dilution
Anti-NeuN	Chemicon (MAB377)	Mouse	1:500
Anti-ChAT	Chemicon (AB144p)	Goat	1:500
Anti-VGluT2	Millipore (AB225)	Guinea pig	1:2500
Anti-VGAT	Synaptic Systems (131 004)	Guinea pig	1:2500
Anti-mouse Alexa Fluor 488	Invitrogen	Donkey	1:500
Anti-goat Alexa Fluor 488/568	Invitrogen	Donkey	1:500
Anti-rabbit Alexa Fluor 555	Invitrogen	Donkey	1:500
Biotinylated anti-guinea pig	Jackson ImmunoResearch	Donkey	1:250

**Table 2 brainsci-10-00824-t002:** Thickness and staining intensity of perineuronal nets around motoneurons.

Groups	Mean Diameter of α-Motoneurons (µm)	Thickness of PNNs (µm)	Mean Intensity of PNNs (Arbitrary Units)
Non-trained	40.35 (±0.50)	2.49 (±0.26)	9690.16 (±1846.7)
Trained	39.86 (±1.10)	3.37 (±0.46)	11,969.11 (±2690.4)

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
