# Peer review of "Effects of Rehabilitation on Perineural Nets and Synaptic Plasticity Following Spinal Cord Transection"

_brainsci, 2020, doi:10.3390/brainsci10110824_

Round 1
Reviewer 1 Report
In this manuscript by Al’joboori et al investigate how the expression of CSPGs, perineuronal nets and markers of presynaptic boutons on motoneurons change following a complete spinal cord injury and after epidural stimulation and locomotion training. They demonstrate that CSPG expression increases with time after SCI and is not affected by ES and training. One of the most interesting observations from this study is that unlike α-motoneurons, γ-motoneurons do not express a PNN, which correlated nicely with their findings on synaptic remodelling. The study investigates two markers of putative synapses onto motor neurons, one excitatory (VGlut2) and one inhibitory (VGat). A higher percentage of synaptic re-organisation was observed on γ-motoneurons that do not express PNNs, which agrees with the current view in the field that PNNs inhibit synaptic plasticity.
The manuscript is comprehensive, very well written and of interest to the field. The experiments are conducted to a high quality with clear aims and conclusions, leading to the identification of novel and valuable findings. I have only very minor comments that need to be addressed before publication:
Minor concerns:
- The first paragraph of the introduction are guidelines from the journal that need to be removed before publication.
- Line 88 – Do you mean stumps instead of “stomps”
- The authors need to co-staining with post-synaptic markers to call these synapses. Alternatively, synapses should be referred as “putative synapses”.
- In the discussion the authors mention the significant role that muscle spindle proprioceptive afferents play in the recovery of function by ES and training after SCI. Did the authors also look at changes to VGlut1 boutons on to motoneurons which would label group Ia proprioceptive afferents?
- Line 212 – I believe the second “21dTR” in the sentence is meant to be “21dNT”.
- Line 222 – Missing a “)” after 3B.
Author Response
Reviewer 1
We very much appreciate the positive feedback from the reviewer, and we have addressed their suggestion as listed below.
- The first paragraph of the introduction are guidelines from the journal that need to be removed before publication.
Reply: Apologies for this mistake, which occurred with the transferring of the text into the template. This has been removed now.
- Line 88 – Do you mean stumps instead of “stomps”
Reply: Yes, apologies for the misspelling. It has now been changed to “stumps”.
- The authors need to co-staining with post-synaptic markers to call these synapses. Alternatively, synapses should be referred as “putative synapses”.
Reply: Yes, we agree that presence of post-synaptic markers would considerably strengthen the certainty of presence of actual synapses. We have followed the reviewer’s suggestions and have changed the text appropriately from “synapses” to “putative synapses” where necessary.
- In the discussion the authors mention the significant role that muscle spindle proprioceptive afferents play in the recovery of function by ES and training after SCI. Did the authors also look at changes to VGlut1 boutons on to motoneurons which would label group Ia proprioceptive afferents?
Reply: The role of Ia afferents in the recovery of locomotion in this model is indeed an important one. However, our focus in this manuscript was to demonstrate possible changes in synaptic boutons and a comparison between inhibitory and excitatory boutons would enrich this discussion. From our previous publications on synaptic plasticity in motor neurones (Ichiyama et al., 2006; Ichiyama et al., 2011), the relationship between F and S type boutons provided significant changes, whereas M-boutons did not change significantly. Therefore, we focused on F (GABAergic/glycinergic) and S (VGluT2+) boutons in this report. Nonetheless, it would have been interesting to investigate VGluT1+ boutons in this model in the future.
- Line 212 – I believe the second “21dTR” in the sentence is meant to be “21dNT”.
Reply: Thank you for the observation. That is indeed the case and it has been corrected.
- Line 222 – Missing a “)” after 3B.
Reply: Again, thank you. This has been added.
Reviewer 2 Report
The authors of this paper acknowledge previous literature that has shown that epidural electrical stimulation (ES) of the lumbar spinal cord promotes functional recovery following SCI. However, they point out that the mechanisms that lead to these improvements have yet to be elucidated. In the present study, the authors aim to identify the effects of ES on perineural nets (PNNs) and synaptic plasticity. They administer ES combined with locomotor training and 5-HT agonists to rats with a SCI at the thoracic level, followed by immunohistochemical analysis. ES and locomotor training did not have any significant effects on PNNs or CSPG expression, however it did induce synaptic remodeling. Interestingly, the motor neurons that did not express PNNs (gamma motor neurons) exhibited a higher degree of synaptic plasticity. Meanwhile, alpha motor neurons expressed PNNs and exhibited reduced synaptic remodeling. Ultimately, this paper suggests that locomotor training (that is aimed to facilitate synaptic remodeling) should be combined with interventions that aim to manipulate CSPGs and PNNs, such as chondroitinase ABC.
However, the introduction does not justify the use of any of the three therapeutic aims. Moreover, there is minimal mention of the context of neuronal injury in the spinal cord, which makes for a poorly justified study. Lack of description of the experimental injury, locomotor training, as well as justification as to why a gel foam was used to separate spinal cord segments when the authors were attempting to promote injury, was not provided. The initial experimental design also did not have all appropriate experimental groups that accounted for each individual treatment (i.e. 5-HT agonist only, locomotor training only, etc.). As such, one or several of these potential therapies could be redundant to the results. The results are displayed very confusingly, where the majority of results only analyze 67DPI.
Comments
- Abstract: Typo in abstract: should be CSPG not CPSG
- Intro: The phrase “A medium negative correlation” is confusing – what do the authors mean by the term “medium”
- Intro: The authors accidentally included the following in their text:
- “The introduction should briefly place the study in a broad context and highlight why it is 32 important. It should define the purpose of the work and its significance. The current state of the 33 research field should be reviewed carefully and key publications cited. Please highlight controversial 34 and diverging hypotheses when necessary. Finally, briefly mention the main aim of the work and 35 highlight the principal conclusions. As far as possible, please keep the introduction comprehensible 36 to scientists outside your particular field of research. References should be numbered in order of 37 appearance and indicated by a numeral or numerals in square brackets, e.g., [1] or [2,3], or [4–6]. See 38 the end of the document for further details on references.”
- Intro: The rationale for investigating PNNs and synaptic plasticity is well explained
- Intro: Evidence in introduction bases a lot of CSPG work in the context of the visual cortex. There may be more relevant model systems to support their work?
- Intro: The authors should explain the rationale for using 5-HT agonists
- This seems like a third, conjunctive treatment and it is also unclear why it is not mentioned in the title of paper.
- Methods The authors did not mention how they distinguished the laminae in their methods – were the laminae divisions approximated? If so, how accurate can these approximations be?
- Methods: What was the daily locomotor training? The training parameters are unclear.
- Methods: Why were additional groups that receive either locomotor training or ES alone not included?
- Methods: Why is Gel foam used to separate to prevent “reconnection of cut ends of spinal cord?” Wouldn’t this impede regeneration that the authors are attempting to promote?
- Discussion: There should be more discussion about why the authors think that training and ES did not have an effect on CSPGs and PNNs
Author Response
Reviewer 2
We very much appreciate the constructive suggestions by the Reviewer. We attempted to address all of the comments as described below and believe the manuscript is improved as a results. Many thanks.
“the introduction does not justify the use of any of the three therapeutic aims”
“there is minimal mention of the context of neuronal injury in the spinal cord, which makes for a poorly justified study”
“Lack of description of the experimental injury, locomotor training, as well as justification as to why a gel foam was used to separate spinal cord segments when the authors were attempting to promote injury, was not provided”
“The initial experimental design also did not have all appropriate experimental groups that accounted for each individual treatment (i.e. 5-HT agonist only, locomotor training only, etc.). As such, one or several of these potential therapies could be redundant to the results”
“The results are displayed very confusingly, where the majority of results only analyze 67DPI”
Reply: We have addressed the reviewer’s concerns above in the specific comments below.
Comments
- Abstract: Typo in abstract: should be CSPG not CPSG
Reply: Thank you for catching that error. It has been corrected.
- Intro: The phrase “A mediumnegative correlation” is confusing – what do the authors mean by the term “medium”
Reply: The terms “small, medium and large correlations” are conventions used in statistical inferences referring to the strength of a correlation. Medium refers to r values > 0.3 and large > 0.5. Our results showed r=0.3 for VGluT2 and r=0.6 for VGat for alpha motoneurones.
- Intro: The authors accidentally included the following in their text:
- “The introduction should briefly place the study in a broad context and highlight why it is 32 important. It should define the purpose of the work and its significance. The current state of the 33 research field should be reviewed carefully and key publications cited. Please highlight controversial 34 and diverging hypotheses when necessary. Finally, briefly mention the main aim of the work and 35 highlight the principal conclusions. As far as possible, please keep the introduction comprehensible 36 to scientists outside your particular field of research. References should be numbered in order of 37 appearance and indicated by a numeral or numerals in square brackets, e.g., [1] or [2,3], or [4–6]. See 38 the end of the document for further details on references.”
Reply: Apologies for this. This was due to transferring the text into the journal’s template. It has now been removed.
- Intro: The rationale for investigating PNNs and synaptic plasticity is well explained
Reply: We assume the reviewer means is NOT well explained? We have added a section introducing the only study showing changes in PNNs after spinal cord injury and rehabilitation which was done in a dorsal column incomplete lesion using chondroitinase as a treatment. We have more clearly justified the need for further investigation to understand how PNNs are regulated with intense locomotor training in a model of complete transection. The fact that we can elicit substantial improvement in locomotor parameters in these completely transected rats with daily locomotor training under ES strongly suggests that plasticity in intraspinal circuits has taken place which facilitates such recovery. We have previously demonstrated significant changes in synaptic composition after a neonatal spinal transection and locomotor training (Ichiyama et al., 2006; Ichiyama et al., 2011). The data presented here are from an adult model of severe injury which avoids developmental issues present in the previous papers.
- Intro: Evidence in introduction bases a lot of CSPG work in the context of the visual cortex. There may be more relevant model systems to support their work?
Reply: We built the evidence regarding the possible role of PNN in the CNS based the current state of the field. The evidence is much stronger and numerous in the brain. We have added one study (Wang et al., 2011) to illustrate the general lack of investigation specifically looking at regulation of PNNs in spinal cord injuries and rehabilitation.
- Intro: The authors should explain the rationale for using 5-HT agonists
- This seems like a third, conjunctive treatment and it is also unclear why it is not mentioned in the title of paper.
Reply: As expertly noted by the reviewer, in this model of complete transection ES alone is not able to produce reliable weight bearing coordinate stepping and a combination with a 5-HT agonist is necessary. However, it was technically and humanely no justifiable to include every single possible control group because of the number of groups necessary to investigate a time series. In our and others previous studies (Courtine et al., 2009; van den Brand et al., 2012) we have clearly demonstrated that completely transected rats receiving locomotor training alone, 5-HT agonist alone, ES alone or ES+5-HT agonist without training are either completely incapable of weight bearing stepping or their ability is severely compromised in comparison to the rats receiving training with both ES and 5-HT agonists. We have changed the title to eliminate the bias provoked by citing epidural stimulation only.
- Methods The authors did not mention how they distinguished the laminae in their methods – were the laminae divisions approximated? If so, how accurate can these approximations be?
Reply: The spinal laminae during surgery were determined based on anatomical landmarks which we have developed over years of experience. Previous dissection demonstrated that in Sprague-Dawley rats of this size spinal segment T9-10 are located under the corresponding vertebral segment T9-10. We locate vertebral segment T13 by following the line of the last rib and count vertebral segments based on this rough landmark. Even though there may be slight anatomical variations between rats, the specific vertebral segments are more precisely determined anatomically by visual inspection (shape, size and position). The curvature of the vertebral column in these rats are a clear characteristic of T10. Our previous dissections have also demonstrated that spinal segment L2 sits between vertebral levels T13 and L1 (which is slightly different from commercially available anatomical atlases), whereas spinal level S1 sits below vertebral level L2. We also verify the location of each epidural electrode during dissections after perfusion and fixation, where the spinal roots are carefully dissected and each segment can be clearly distinguished. We did not note any major deviations in placement of electrodes in this series of studies.
- Methods: What was the daily locomotor training? The training parameters are unclear.
Reply: Because the parameters of training have been described extensively previously and the current study did not deviate from those, we only presented a summary of major parameters. We noted in the beginning of Methods that such descriptions have already been previously published. We have added such sentence in the Locomotor Training section as well.
- Methods: Why were additional groups that receive either locomotor training or ES alone not included?
Reply: This issue was addressed in a reply above.
- Methods: Why is Gel foam used to separate to prevent “reconnection of cut ends of spinal cord?” Wouldn’t this impede regeneration that the authors are attempting to promote?
Reply: The main objective of these studies is to understand the plasticity that occurs within spinal circuits caudal to the complete transection which underlie the functional recovery observed with rehabilitation. We and others have demonstrated that no axonal regeneration is observed in complete transections of the spinal cord in adult rats. Our objective was not to induce such regeneration with our interventions, but rather to understand the plasticity mechanisms within the lumbar spinal cord that lead to improvements in stepping ability.
- Discussion: There should be more discussion about why the authors think that training and ES did not have an effect on CSPGs and PNNs
Reply: We have clarified this session in the text, but the issue of lack of investigations in this area remains. We have gone so far as to speculate that the complete transection may account for such discrepancy, because they remain paralysed in their home cages, unlike our previous two studies using an incomplete cervical lesion and intact rats. We are currently finalizing another paper in which we applied ES and training in a severe contusion injury model where we do observe an increase in CSPG expression caudal to the lesion in the trained rats, which further corroborates our hypothesis of a link between levels of activity and expression of CSPGs.
References
Courtine G, Gerasimenko Y, van den Brand R, Yew A, Musienko P, Zhong H, Song B, Ao Y, Ichiyama RM, Lavrov I, Roy RR, Sofroniew MV, Edgerton VR (2009) Transformation of nonfunctional spinal circuits into functional states after the loss of brain input. Nature neuroscience 12:1333-1342.
Ichiyama RM, Broman J, Edgerton VR, Havton LA (2006) Ultrastructural synaptic features differ between alpha- and gamma-motoneurons innervating the tibialis anterior muscle in the rat. The Journal of comparative neurology 499:306-315.
Ichiyama RM, Broman J, Roy RR, Zhong H, Edgerton VR, Havton LA (2011) Locomotor training maintains normal inhibitory influence on both alpha- and gamma-motoneurons after neonatal spinal cord transection. The Journal of neuroscience : the official journal of the Society for Neuroscience 31:26-33.
van den Brand R, Heutschi J, Barraud Q, DiGiovanna J, Bartholdi K, Huerlimann M, Friedli L, Vollenweider I, Moraud EM, Duis S, Dominici N, Micera S, Musienko P, Courtine G (2012) Restoring voluntary control of locomotion after paralyzing spinal cord injury. Science 336:1182-1185.
Wang D, Ichiyama RM, Zhao R, Andrews MR, Fawcett JW (2011) Chondroitinase combined with rehabilitation promotes recovery of forelimb function in rats with chronic spinal cord injury. The Journal of neuroscience : the official journal of the Society for Neuroscience 31:9332-9344.